# Impacts of land use on soil carbon, nitrogen, and phosphorus in the Eastern Qilian Mountains

Shizhen Xu[1], Chunli Wang[1,2]*, Junju Zhou [1,2,3], Haihua Shan[1], Bingxing Li[1], Wei Shi[1], Dongxia Zhang[1], Guofeng Zhu[1,2], Xuemei Yang[4], Wei Wei[1,2], Haiyan Ma[5]

1 College of Geography and Environmental Science, Northwest Normal University, Lanzhou, China, 2 Key Laboratory of Resource Environment and Sustainable Development of Oasis, Lanzhou, Gansu, China, 3 Gansu Engineering Research Center of Land Utilization and Comprehension Consolidation, Lanzhou, China, 4 Tourism school, Lanzhou University of Arts and Science, Lanzhou, China, 5 Dang River Basin Water Resources Management Bureau, Jiuquan, Gansu, China

* yzh_su@163.com

## Abstract

The dynamics and balance of soil carbon, nitrogen, and phosphorus significantly affect soil microbial activity and plants' nutrient absorption and utilization. Under-standing how different land-use types and climate fluctuations influence soil carbon, nitrogen, and phosphorus forms the basis for regional land-use optimization, sci-entific management, and enhancement of ecosystem service functions.This paper continuously collected soil samples from forestland, grassland, and cropland in the Binggou River Basin on the northern slope of the eastern section of the Qilian Mountains during 2018–2019. It analyzed variation patterns of soil carbon, nitrogen, and phosphorus across different land-use types in the study area and explored the impacts of land-use types and seasonal climate fluctuations on these soil elements, aiming to provide a scientific basis for soil management. Results showed that during 2018–2019, the average organic carbon content in forestland soil (48.82 g/kg) was higher than that in grassland (38.32 g/kg) and cropland (30.80 g/kg). Forestland soil had slightly higher average total nitrogen (TN) content than cropland, with grassland showing the lowest. Soil organic carbon (SOC) and TN contents in forestland and grassland were lowest in summer, while grassland's total nitrogen content peaked in summer. The average total phosphorus (TP) content in cropland soil was slightly higher than in forestland and significantly higher than in grassland. The weaker carbon-sequestration capacity of cropland soil resulted in notably lower C:N (10.13) and C:P (47.38) ratios compared to forestland and grassland. Soil C:P and N:P ratios in grassland and cropland showed relatively small seasonal fluctuations, whereas those in forestland fluctuated more drastically, reaching the highest values in autumn. Excessively high C:P reduced soil phosphorus effectiveness. Regarding the soil vertical profile, SOC and TN contents in forestland and grassland both decreased with increasing soil depth, while the three nutrients in cropland soil exhibited more

**Data availability statement:** All relevant data are within the manuscript and its Supporting Information files.

**Funding:** This research was supported by the National Natural Science Foundation of China under grants JZ (42361005), GZ (41867030), XY (32060373), and CW (23SLK086). No additional external funding was received for this study.

**Competing interests:** Declarations Conficts of interest No conflict of interest exits in the submission of this manuscript, and manuscript is approved by all authors for publication. I would like to declare on behalf of my co-authors that the work described was original research that has not been published previously, and not under consideration for publication elsewhere, in whole or in part. All the authors listed have approved the manuscript that is enclosed.

complex vertical variation characteristics. Overall, converting forestland and grassland to cropland significantly reduced soil carbon-sequestration capacity, shifting them from "carbon sinks" to "carbon sources" and increasing carbon emission risks. Changes in organic matter input, tillage practices, and chemical fertilizer use altered the vertical nutrient variation patterns in cropland soil.

## 1. Introduction

Soil provides the material basis and essential conditions for human survival, production, and development. It is closely linked to population, energy, environment, and ecology through the soil system [1]. As a vital component of the ecosystem, soil C, N, and P content are closely related to litter decomposition rates, soil microorganism populations, and the long-term accumulation of soil organic carbon and nutrients [2,3]. These factors are key drivers of the biogeochemical cycle in terrestrial ecosystems [4,5]. C, N, and P cycles are interconnected [6], and their changes during biogeochemical cycles influence soil C balance, determining the C source and sink functions of terrestrial ecosystems [7]. Additionally, N and P are the primary limiting factors for plant growth [8,9]. Therefore, studying the balance of soil C, N, and P is crucial for understanding nutrient limitation and cycling patterns in terrestrial ecosystems [10,11]. Soil C:N:P stoichiometric relationships are recognized as valuable indicators of soil nutrient status and processes across ecosystems. Specifically, C:N and C:P ratios can reflect organic matter decomposition rates, nutrient mineralization or fixation, and plant nutrient limitations. Therefore, an integrated analysis of soil nutrient content and stoichiometric ratios provides a theoretical basis for balanced land use [12].

Soil nutrients and stoichiometry vary based on soil type, physicochemical properties, environmental factors, and plant species [13,14]. Differences in the mineralization, transport, uptake, and utilization of C, N, and P elements occur in soils under different land use practices [13]. Plant properties and human activities significantly impact soil C, N, and P contents, as well as stoichiometry [15,16]. In general, compared to grassland, forest land has higher SOC and TN, with no significant difference in TP content [17,18].

However, the effect of land use type on soil nutrient content varies across regions due to factors such as climate and topography. For example, in the Shuanglong catchment of Dianchi Lake in southwest China, SOC and TN contents in grassland were higher than those in forestland at depths of 0–40 cm [19]. In the high mountains of the northeastern Tibetan Plateau, local climate, vegetation variation, and soil development played a key role in soil nutrient stoichiometry due to unique climatic conditions, vegetation succession, and soil development. Local climate, vegetation variation, and soil development played a key role in soil nutrient stoichiometry [20]. Global climate change, coupled with drastic human activities, has altered land use/cover and soil properties, leading to more complex regional variability in soil nutrient content. For example, in a typical agricultural-pastoral region of northwest China,

converting pastureland to farmland led to significant losses of SOC, TN, and TP in both topsoil and subsoil, with subsoil experiencing a higher reduction rate than topsoil. After converting farmland to forestland, SOC, TN, and TP contents increased significantly [13,21]. In the Loess Plateau of China, the policy of converting farmland to forest and grassland has led to extensive conversion of agricultural land to forestland and grassland, significantly increasing SOC, TN, and TP storage in shallow soils [22–24]. The southern Songnen Plain is a typical soil salinization area in China. Different land use types are the primary factors influencing soil nutrient content and stoichiometric ratios, while re-vegetation of cropland improves soil nutrients. Its effect on soil stoichiometric ratios is more pronounced in the surface soil (0–20 cm) compared to deeper soil (20–50 cm) [25]. This indicates that tillage causes significant loss of soil nutrients and microbial populations, whereas converting farmland to forest or grassland effectively promotes microbial diversity, increases soil nutrient content, and improves soil quality [13]. In contrast, deforestation, overgrazing, and intensive tillage can reduce SOC, TN, and TP content, leading to land degradation and decreased soil fertility. Therefore, studying the impact of land use changes in different regions on soil carbon, nitrogen, and phosphorus cycles, as well as exploring the underlying mechanisms, can help optimize regional ecological protection and restoration policies.

The Qilian Mountains, located at the northern edge of the Tibetan Plateau, are a sensitive area to global change and play a crucial role in maintaining regional ecological security as a national nature reserve. This semi-arid mountain ecosystem, with elevations ranging from 2,000–5,500 meters, is characterized by complex topography and steep elevation gradients, resulting in considerable variability in vegetation and soil patterns [26]. In recent decades, global climate change has had a significant impact on the Qilian Mountain ecosystem. In addition, the reclamation of forests and grasslands by local residents further damaged the ecosystem. From 2009 to 2019, the farmland area increased by 1.0278 km²; the forestland area increased by 4.4199 km²; the grassland area decreased by 6.246 km²; the water body area decreased by 0.4761 km²; the unused land area increased by 1.2744 km²; and the Build-upland area remained unchanged [27]. Although China has implemented a series of ecological restoration projects since 2000, including natural forest protection, the construction of ecological public welfare forests, and the return of farmland to forest (or grassland) to protect and restore the degraded ecosystem [28], some of the reclaimed farmland has yet to be converted back to grassland or forest. Changes in land use types have altered ecosystem services and impacted the processes of soil nutrient retention and cycling. Despite progress, few studies have focused on the impact of land use change on the soil carbon, nitrogen, and phosphorus cycles in the eastern Qilian Mountains. This lack of understanding of the impact mechanisms has become a bottleneck for optimizing ecological conservation and restoration measures. This study focuses on the Binggou River basin on the northern slope of the eastern Qilian Mountains. It analyzes the impact of three land use types—forest land, grassland, and cropland—on soil carbon, nitrogen, and phosphorus storage and ecological stoichiometry. Additionally, it explores how land use changes influence soil carbon, nitrogen, and phosphorus, Clarifying the effects of land-use change and seasonal variation on the state of soil carbon, nitrogen and phosphorus balance, and provides theoretical support for ecological conservation and restoration efforts in the Qilian Mountains. The innovation and significance of this paper are mainly reflected in the following two aspects: First, a comparative study was carried out on the changes of soil carbon, nitrogen and phosphorus in different land use types in the Qilian Mountains, and the impact of land use type transformation on soil carbon sequeeration capacity and soil nutrient balance was clarified. Second, the research results have certain guiding significance for ecological restoration in Qilian Mountains.

## 2. Materials and methods

### 2.1. Description of the study area

The Binggou River Basin is situated in the northeastern Qilian Mountains, with coordinates of 37°34′N to 37°47′N latitude and 102°10′E to 102°31′E longitude. The Binggou River watershed consists of two tributaries: the Binggou River and the Nancha River, and the Binggou River watershed is 45 km long with a watershed area of 335 km². The terrain is

characterized by high elevations in the southwest and lower elevations in the northeast, with steep alpine valleys. The climate is temperate continental, with cold, long winters, cool summers, and short spring and autumn seasons. The basin has an annual average temperature of 3.5°C, with annual precipitation ranging from 400 to 600 mm and annual evaporation ranging from 800 to 1,000 mm. July is the hottest month, with a monthly average temperature of 18.2°C. Precipitation shows significant seasonal variation, with the majority occurring in summer. In July and August alone, precipitation accounts for over 50% of the annual total.

The Binggou River Basin spans elevations from 1,947 m to 4,560 m, with a well-developed vertical zonation. Below 2,500 m, the area is predominantly desert, while between 2,500 m and 3,000 m, grassland and forestland are found. Along the riverbanks, some of the grassland and forestland have been converted into farmland. Above 3,000 m, alpine meadows and glaciers dominate. The soil types from downstream to upstream include desert soil, mountain gray soil, mountain chestnut soil, mountain gray cinnamon soil, and meadow soil.

In the *Picea crassifolia* forest, the canopy density of the arbor layer was relatively high, which led to insufficient light beneath the forest. Consequently, shrubs exhibit poor growth and development, resulting in relatively low species diversity. This forest community was mainly composed of deciduous shrubs, such as *Potentilla fruticosa* and *Potentilla glabra* from the Rosaceae family. The herb layer predominantly comprised typical alpine meadow plants, including *Carex sp.*, *Polygonum viviparum*, and *Pedicularis sp*. The bryophyte layer was well – developed, primarily consisting of *Abietinella abietina, Hypnum cupressiforme, Mnium cuspidatum*, etc. The grassland community featured a simple structure, mainly composed of species like *Stipa spp., Agropyron cristatum, and Oxytropis spp*. Farmland in the Ice Gully River Basin was mainly located along the river and was the result of the development of riparian forestland and grassland. The arable land at the sampling site had been cultivated for 32 years and was mainly planted with wheat and potatoes.

## 2.2. Soil sampling and analyses

### 2.2.1. Determination of sampling points.
Fig 1 shows that in the Binggou River Basin, farmland is primarily found along the riverbanks at elevations of 2,400−2,700 m, grassland is mainly located between 2,500 and 3,000 m, and forestland is predominantly found at elevations between 2,200 and 2,800 m. To conduct a comparative study on the effects of land use types on soil carbon, nitrogen, and phosphorus cycles under similar climate and terrain conditions, this study considered the spatial distribution characteristics of each land use type and selected the elevation range (2,500−2,700 m), where all three land use types overlap, as the optimal sampling altitude. Additionally, considering the location of the weather station (102°20'63"E, 37°42'09"N), a field investigation was conducted from 2017 to 2018 to finalize the sampling point at an elevation of 2,539 m (102°20'41"E, 37°41'97"N), with a horizontal distance of 560 m from the meteorological station.

### 2.2.2. Soil sampling.
Sampling was conducted during the vegetation growth period from May to October in 2018 and 2019 (Fig 2). The sampling process is as follows: one independent sample plot (20m × 20m) was selected for each land type, and five sample plots were established using the five-point sampling method, each measuring 3m × 3m (Fig 1). Litter was removed from the ground prior to soil sample collection. Soil samples were collected using a soil borehole sampler with an inner diameter of 5 cm. Samples were collected from the following depths: 0–10 cm, 10–20 cm, 20–30 cm, 30–40 cm, 40–50 cm, and 50–60 cm. The soil samples from each layer of the five quadrants in each independent plot were mixed to form a composite sample. The fresh soil samples were then placed into sampling bags, labeled with the sampling location, time, and depth, and transported to the laboratory. In total, 1080 soil samples were collected (3 plots × 5 squares × 6 depths × 6 months × 2 years = 1080 samples).

## 2.3. Experimental scheme

Soil samples were air-dried for a fortnight in a cool, ventilated environment. Plant debris and gravel were removed from the soil samples prior to soil splitting. The air-dried soil samples were ground and passed through a 100 mesh sieve to

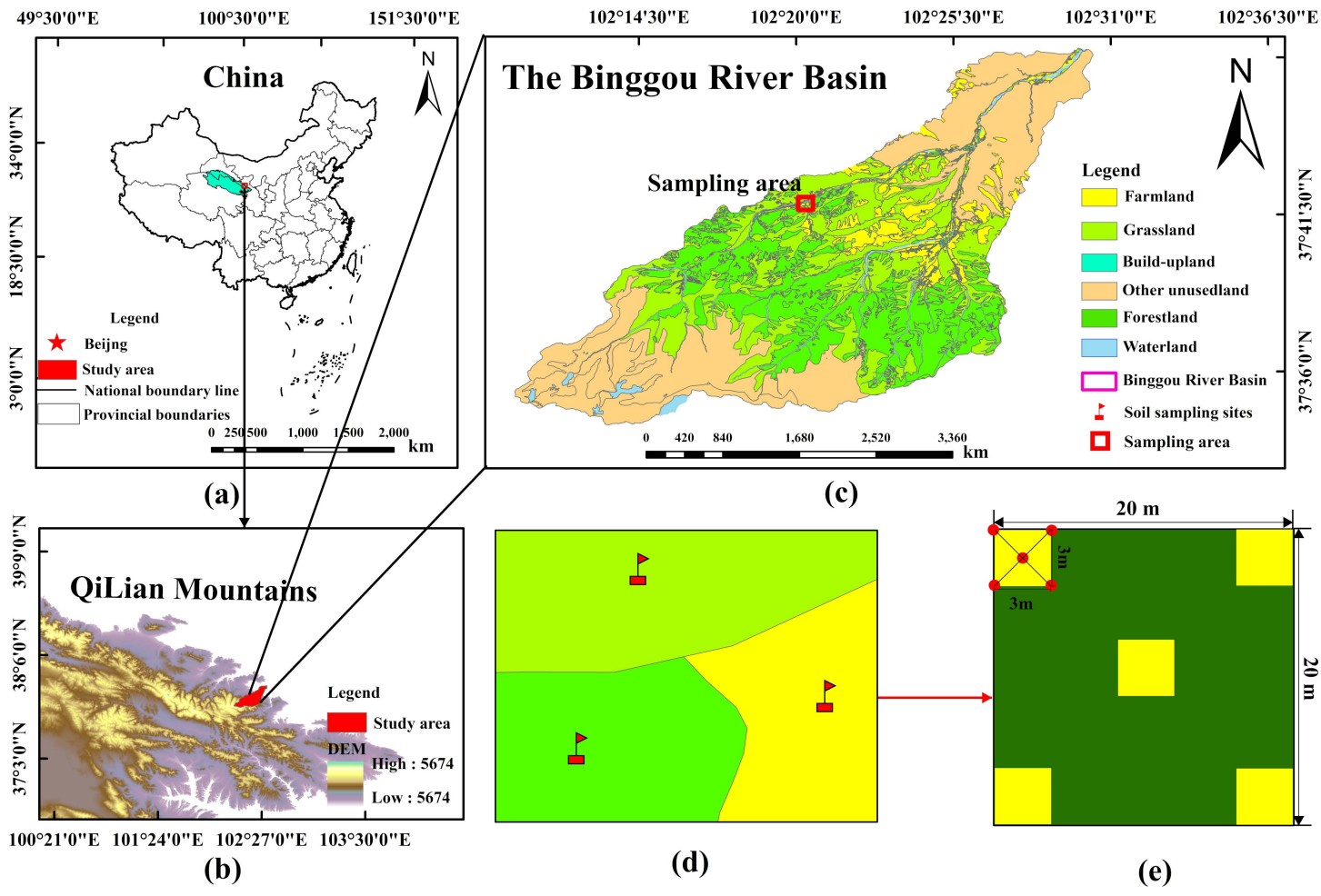

**Fig 1. The sampling location and process of the Binggou River Basin on the Qilian Mountains.**

remove coarse fragments and debris for the determination of soil organic carbon (SOC), soil total nitrogen (TN), and soil total phosphorus (TP).

**2.3.1. Determination of SOC.** Weigh 0.2 g of the soil sample and place it into 10 mL of potassium dichromate sulfuric acid solution. Heat at 180°C for 7 minutes, then remove and cool. After cooling, transfer the liquid from the digestion tube into a 250 mL Erlenmeyer flask and add 60–70 mL of distilled water. Add four drops of phenanthroline indicator and titrate with ferrous sulfate solution, observing the color change. Record the volume of titrant consumed. Prepare a blank for each set of samples (the blank should be identical to the sample, except for the absence of soil). Since ferrous sulfate is prone to oxidation, calibrate the potassium dichromate reference solution daily to determine the concentration of the ferrous sulfate solution. The soil organic carbon content was calculated as follows:

$$SOC = \frac{(V_0 - V) \times C_2\,(Fe_2SO_4) \times 0.003 \times 1.1}{m} \times 1000$$

$V_0$: number of ferrous sulfate consumed by blank; V: number of ferrous sulfate consumed by titrated soil samples; m: dried soil sample weight; 0.003: number of 1 milligram equivalent carbon; 1.1: oxidation correction coefficient.

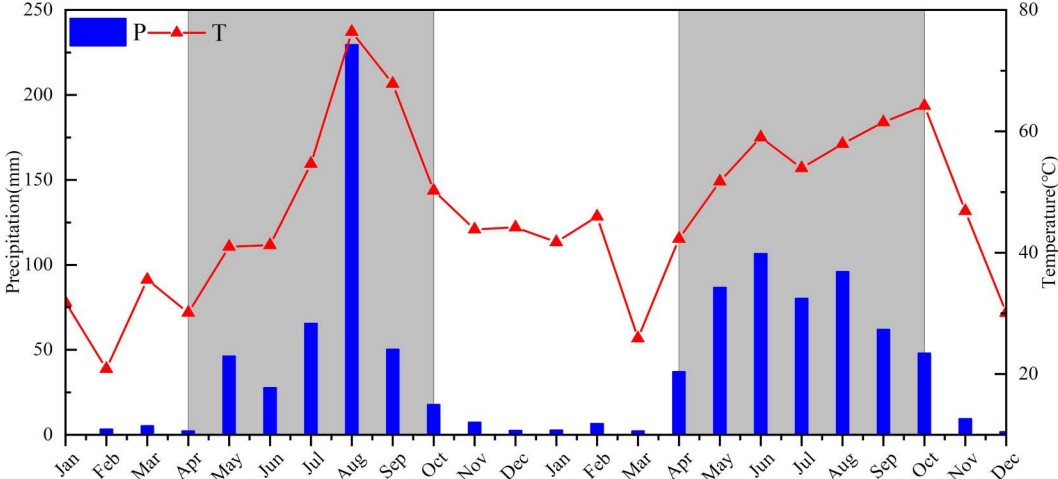

**Fig 2. Distribution of temperature and precipitation in the study area (2018-2019).**

**2.3.2. Determination of total soil nitrogen.** Soil Sample Digestion: Weigh 0.5g of air-dried soil, previously passed through a 100-mesh sieve, and place it at the bottom of a clean digestion tube. Carefully place the sample at the bottom of a clean digestion tube, add a few drops of distilled water to moisten, and then add 1g of mixed catalyst. Add 5mL of concentrated sulfuric acid and gently swirl the tube to mix. Cover the digestion tube with a small funnel. Set the digestion instrument to 370°C and a digestion time of approximately 4 hours, then press "Start" to begin heating. Continue heating until the solution turns clear and light blue-green, which should take about 4 hours. After digestion, remove the tube and allow it to cool. Add 20mL of distilled water to dissolve the digestion liquid, and transfer it to a 100mL volumetric flask. Rinse the digestion tube several times with distilled water and pour the rinses into the volumetric flask. Finally, make up the volume to the mark, mix thoroughly, and allow it to settle (for the blank sample, follow the same steps but omit the soil). The next day, take 5mL of the clear upper liquid and transfer it to a 25mL volumetric flask. Fill the flask to the mark, mix well, and the solution is ready for analysis.

Soil Sample Measurement: The measurement is performed using the Smartchem 200 automated chemical analyzer, branded AMS Alliance, Italy.

**2.3.3. Determination of total phosphorus in soil.** Soil Sample Digestion: Accurately weigh 0.5g of air-dried soil, passed through a 100-mesh sieve, and place it into a 100mL digestion tube. Add 5mL of concentrated sulfuric acid and swirl to mix. Next, add 10 drops of perchloric acid, swirl again, and cover the tube with a small funnel. Heat the tube in a digestion furnace, gradually increasing the temperature. When the temperature reaches approximately 280°C, the solution will turn white and become transparent. Continue heating until the temperature reaches 370°C, then digest for 1 hour. The total digestion time is approximately 2 hours. After digestion, allow the digestion tube and solution to cool to room temperature. Rinse the remaining digestion solution from the tube several times with distilled water, transferring all liquid to a 100mL volumetric flask. Dilute the solution to 100mL with distilled water, filter or clarify, and the sample is ready for analysis (the blank sample follows the same procedure, excluding the addition of soil). The next day, take 5mL of the clear upper liquid and transfer it to a 25mL volumetric flask. Fill to the mark, mix well, and the sample is ready for analysis.

Soil Sample Measurement: The measurement is performed using the Smartchem 200 automated chemical analyzer, branded AMS Alliance, Italy.

## 2.4. Other data

During the sampling period, temperature, precipitation, and other meteorological elements at the sampling points were monitored by an automatic weather station, with accuracies of 0.01°C and 0.2mm, and a recording

interval of 15 minutes. The daily average temperature and daily precipitation data for the sampling points were calculated (Fig 2).

Land use data were obtained from Landsat-8 OLI images from August 2019.Using an online map and ArcGIS 10.2 software, watershed land use was classified through manual visual interpretation. The study area was classified into six land use types: farmland, forest land, grassland, water bodies, built-up land, and unused land.

## 2.5. Statistical analysis

The time and vertical variation characteristics of SOC, TN and TP contents were analyzed by descriptive statistical method. Data were analysed using SPSS 22.0 software, and all data were expressed as mean±standard deviation *(P<0.05)* using one-way ANOVA, multiple independent samples rank-sum test (Kruskal-WallisH test) and LSD post-hoc test for the content of SOC, TN, and TP as well as the content of C/N, C/P, and N/P in the different land-use modes. significicance level tests *(P<0.05)* were performed.

## 3. Results

### 3.1. Changes in SOC, TN and TP

**3.1.1. Time change.** From Fig 3, it is observed that the SOC content is highest in forestland (49.25±6.65 g/kg), followed by grassland (38.01±5.7 g/kg), and lowest in farmland (33.21±5.46 g/kg). Statistical tests (ANOVA) revealed significant differences in SOC among land use types *(P<0.05)*.The conversion of forestland-and grassland into ploughed land reduces SOC content by an average of 16.04 g/kg and 4.8 g/kg, respectively. The annual changes in SOC content in forestland, grassland, and farmland were similar, displaying a "U" shape with a minimum in summer. Compared to 2018, the SOC content in 2019 exhibited more stable annual variation, primarily due to larger seasonal fluctuations in monthly precipitation in 2018, whereas in 2019, the fluctuations were smaller (Fig 2).

No significant difference was observed in the average soil TN content among forestland, grassland, and farmland *(P>0.05)*, though TN content fluctuated considerably throughout the year. The TN content in forestland fluctuated considerably during the growing period, similar to that of SOC. The TN content in farmland exhibited minimal annual variation, fluctuating around 3.00 g/kg, similar to that of forestland overall. However, the fluctuation of TN content in grassland from May to October was largely opposite to that in forestland, peaking in summer (3.27 g/kg).

The TP concentration in forestland was relatively low (0.46±0.09 g/kg), remaining stable in spring and summer but decreasing significantly in autumn. The TP content in grassland soil was slightly higher than in forestland (0.59±0.04 g/kg), with relatively stable fluctuations, reaching its lowest value (0.52 g/kg) in July. The soil TP content in farmland was higher than in grassland and forestland (average value: 0.65±0.07 g/kg), exhibiting higher levels in spring and autumn, and lower levels in summer, similar to the fluctuation trend in grassland soil TP.Statistical results (ANOVA) indicated significant differences in TP among land use types *(P<0.05))*.

Overall, there was little difference in TN content among the three land types, with forestland having the highest SOC content and the lowest TP content. In summer, forestland exhibited the lowest SOC and TN and the highest TP, whereas grassland had the lowest SOC and TP, but the highest TN. The lowest values of SOC, TN, and TP in farmland occurred between July and August. The TN and TP contents in forestland and grassland fluctuated in opposite directions.

**3.1.2. Vertical change.** The vertical changes in SOC, TN, and TP contents in forestland soil showed strong consistency, with a clear decreasing trend from the surface to the deeper layers. The decrease rates of SOC and TN were highest in the 0–30 cm depth, while SOC and TN content stabilized below 40 cm at 40.02±1.85 g/kg and 2.75±0.57 g/kg, respectively. The TP content in forestland soil showed the highest decrease in the 10–40 cm layer (Fig 4).

The vertical changes in SOC and TN contents in grassland soil were consistent, both showing a decreasing trend. The highest decrease occurred at 0–30 cm, while content stabilized at 34.92±0.95 g/kg and 2.36±0.41 g/kg at depths of 30–60 cm. However, the TP content in grassland soil initially decreased and then increased from the surface to deeper

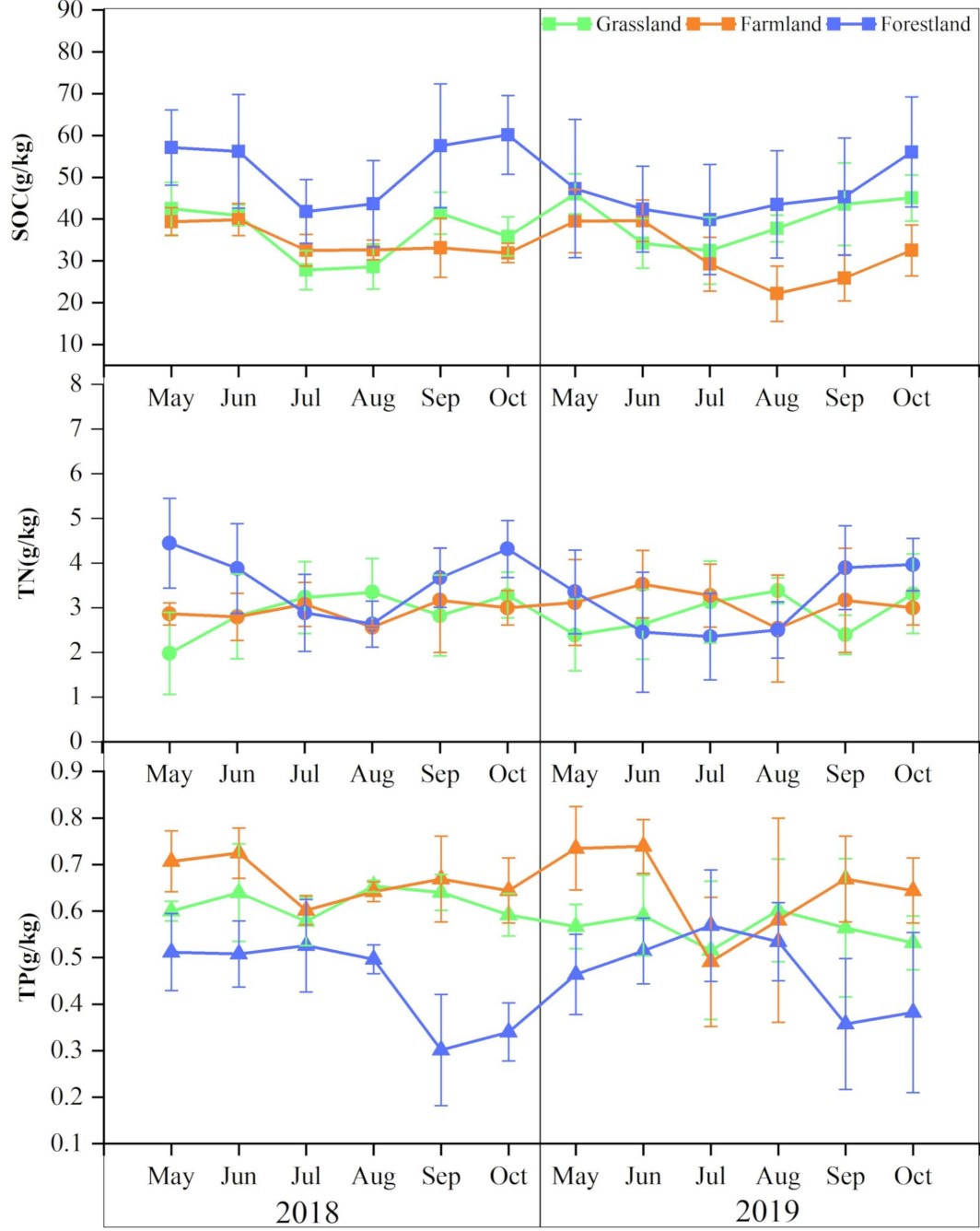

**Fig 3. Concentrations of C, N, and P in time change.**

layers, reaching its minimum of 0.54 g/kg at 20–30 cm. In farmland, SOC, TN, and TP all exhibited "S" type changes in the vertical soil profile.

In general, SOC and TN in natural forest and grassland soils primarily result from the decomposition of plant and animal residues. As soil depth increases, the input of organic matter gradually decreases, leading to a reduction in SOC

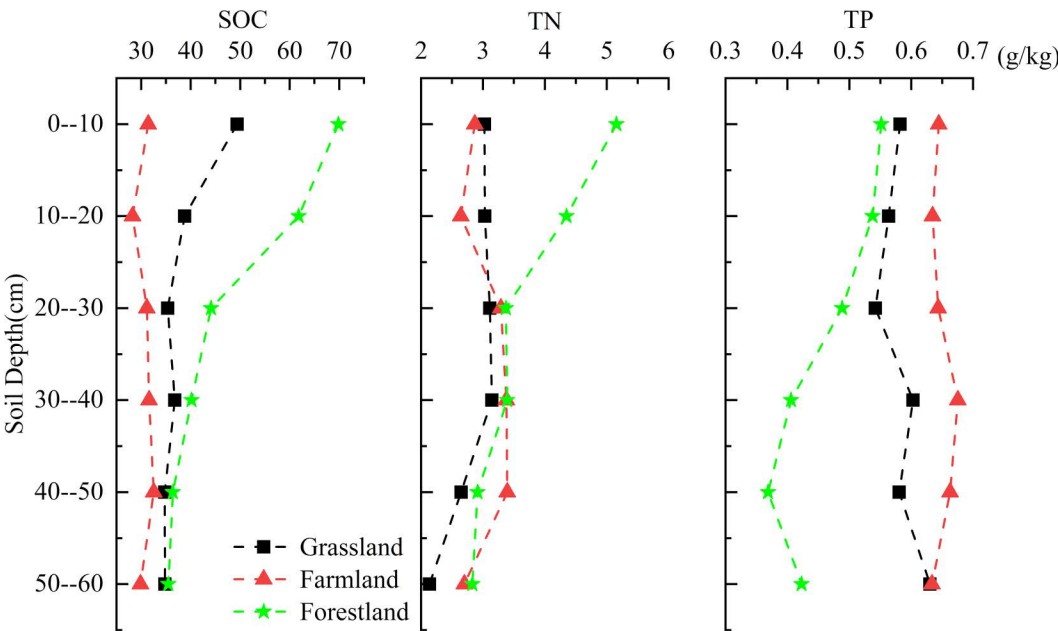

**Fig 4. Concentrations of SOC, TN, and TP in vertical feature.**

and TN content. The trend of TP changes in grassland soil differs from that of SOC and TN, closely linked to the source of TP. In addition to organic matter inputs, soil parent material continuously replenishes phosphorus in deeper layers. Consequently, the vertical distribution of phosphorus in the soil profile remains relatively stable, but in forestland soils, the thicker soil layer results in a weak recharge process, causing rapid depletion of TP from the surface downward. Farmland is significantly influenced by human activities, such as tillage, sowing, and fertilization, which disrupt the natural vertical distribution of soil nutrients, leading to a more uniform nutrient distribution across layers.

There were differences in the effects of land use type and soil depth on SOC, TN, and TP contents. Generally, significant differences in SOC content existed between forest land and grassland, which passed the 0.05 significance level test *(P<0.05)*. Additionally, there were significant differences in SOC content between the top soil layer (0–10 cm) and the remaining layers. In contrast, the SOC content in arable land did not pass the 0.05 significance level test *(P>0.05)*, and no significant differences were observed in SOC content across different soil depths. Regarding TN content, only significant differences were found between forest land (as a whole) and different soil depths *(P<0.05)*, while no significant differences existed between cultivated land/grassland (as a whole) and soil depths *(P>0.05)*. Different soil depths significantly affected TN content, with the overall TN mean values showing a decreasing trend as depth increased. The 0–10 cm depth significantly differed from most other depths, and notable differences were also observed in specific depth intervals (e.g., 20–30 cm, 30–40 cm, 40–50 cm, 50–30 cm). TP content behaved similarly to TN. Significant differences were only observed between forest land and different soil depths *(P<0.05)*, but no significant differences existed between cultivated land/grassland (as a whole) and soil depths *(P>0.05)*. Soil depth significantly influenced TP content, following a linear trend. The mean TP contents in the top soil layers (0–10 cm and 10–20 cm) differed significantly from those in deeper layers (30–40 cm and 40–50 cm).

## 3.2. Changes in soil ecological stoichiometry

**3.2.1. Time change.** Soil C:N ratio is an important indicator of the balance between carbon and nitrogen in soil. When the C:N ratio is between 20 and 30, soil mineralization and assimilation are balanced. The C:N ratio in grassland

is lower in summer (5–15) and higher in spring and autumn (10–25). The C:N ratio in forestland ranges from 10 to 25, fluctuating minimally, with higher values in summer and lower values in spring and autumn, indicating a balance between soil mineralization and assimilation. Similarly, the soil C:N ratio in farmland also fluctuates minimally, ranging from 5 to 15 (Fig 5). The C:N ratio in grassland and farmland is relatively low in summer, promoting rapid nitrogen mineralization and release, which benefits plant absorption. In summer, the C:N ratio in grassland is lower, while in forestland it is higher. In spring and autumn, the C:N ratio in grassland is higher, while in forestland it is lower, demonstrating the opposite trend. The lower the C:N ratio, the faster nitrogen is released into the soil for crop use [29]. This indicates that nitrogen content in grassland is more abundant in summer, benefiting herbaceous plant absorption. In forestland, nitrogen content is higher,

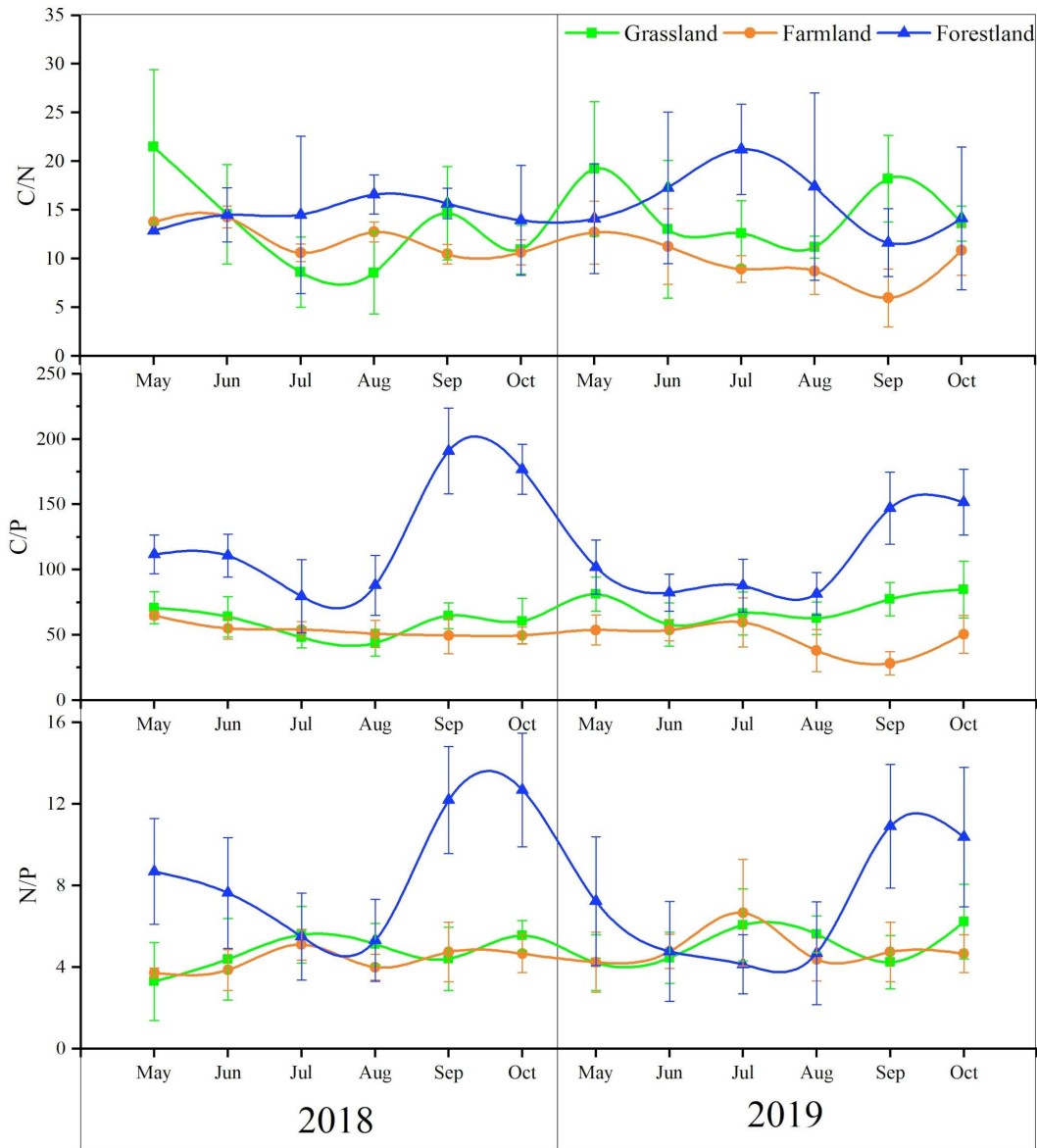

**Fig 5. The ecological stoichiometry of soil in time feature.**

aiding tree growth in spring and autumn. The C:N ratio in farmland remains low, which benefits crop nutrient absorption, likely due to the addition of nitrogen fertilizers.

For the three land types, the soil C:N ratio in farmland was lower than that in grassland and forestland, with a significant difference compared to forestland in 2019 (except in May and September). In summer, soil C:N in grassland was significantly lower than in forestland, while in spring and autumn, it was higher than in forestland. The overall fluctuation of the C:N ratio in grassland was greater than in forestland and farmland.

The soil C:P ratio is an important indicator of phosphorus mineralization capacity and a potential measure of soil organic matter's ability to release or absorb phosphorus from the environment [30]. In forestland, the soil C:P ratio is lower in spring and summer, but higher in autumn. The C:P ratio fluctuates significantly overall, ranging from 50 to 120 (Fig 5). This study analyzed that soil phosphorus content is high in spring and summer, allowing plants to fully utilize the phosphorus required for growth. In autumn, soil phosphorus content is deficient, which limits the decomposition and mineralization capacity of soil microorganisms and hinders nutrient absorption by vegetation. The C:P ratio in grassland and farmland was lower than in forestland, ranging from 45–80 and 30–65, respectively, with steady fluctuations. This indicates that soil phosphorus in grassland and farmland is sufficient, which supports the growth of vegetation and crops. Therefore, phosphorus deficiency is most prominent in forestland during autumn.

The soil N:P ratio can indicate the supply of soil nutrients during plant growth and help determine the threshold of limiting nutrients [31]. Generally, the soil N:P ratio fluctuates within a specific range as vegetation grows. Changes in the N:P ratio suggest that plant growth may be limited by either nitrogen or phosphorus. The N:P ratio in grassland and farmland remained stable, with similar trends showing lower values in spring and autumn and higher values in summer. In forestland, the N:P ratio fluctuated minimally in spring and summer, but was significantly higher than that of grassland and farmland in spring, slightly lower in summer of 2018, and significantly lower in summer of 2019. Soil N:P in forestland increased in autumn, reaching a maximum of 12.18, significantly higher than that of grassland and farmland. During the summer, vegetation is in the rapid growth phase, with strong carbon sequestration capacity in forestland. Nitrogen absorption is not limited by carbon, allowing for full utilization by vegetation. Grassland and farmland have weaker carbon sequestration capacity, with nitrogen consumption limited by carbon, leading to nitrogen accumulation in the soil that cannot be fully absorbed by plants. During the investigation of the study area, it was found that grazing existed in the grassland, and nitrogen fertiliser would be used in the arable farming process, which increased the input of nitrogen, but the carbon sequestration capacity of the grassland and the arable land was weaker (Fig 3), and the consumption of nitrogen was limited by carbon, which couldn't be absorbed by the plants sufficiently, and it was accumulated in the soil, which resulted in the TN content of the soil of the grassland was higher than that of its month of July in summer, and the TN content of the soil of the arable land decreased, but to a lesser extent, in July (Fig 3). month was reduced, but the reduction was small (Fig 3). On the other hand, the phosphorus content of grassland and arable soils decreased significantly in summer, especially in July, resulting in higher soil N:P ratios in summer.

In summary, the ecological stoichiometric ratios across the three land use types showed minimal fluctuations in the C:N ratio, indicating that soil C:N ratios are relatively stable and less influenced by land use/cover types under consistent hydrothermal conditions within the same area. Soil C:N and N:P were significantly higher in soils in autumn than in other seasons, and excessively high C:N (> 175:1) reduced soil phosphorus effectiveness.

**3.2.2. Vertical change.** Vertically, the C:N ratios of the three land use/cover types exhibited a pattern of "decreasing first and then increasing," with the overall change being relatively small (Fig 6). The minimum C:N ratio for forest land (11.84) occurred in the 30–40 cm soil layer, while the minimum C:N ratio for grassland (11.37) was found in the 20–30 cm layer. The C:N ratio of farmland fluctuated around 10.18, and the C:N ratio of soil surface organic matter under conventional tillage ranged from 8 to 15, with an average of 10–12 [32].

As soil depth increased, the C:P ratio of the three soil types gradually decreased, with a more pronounced decrease in the surface layer compared to the middle and lower layers. The N:P ratio in forest land showed a decreasing trend

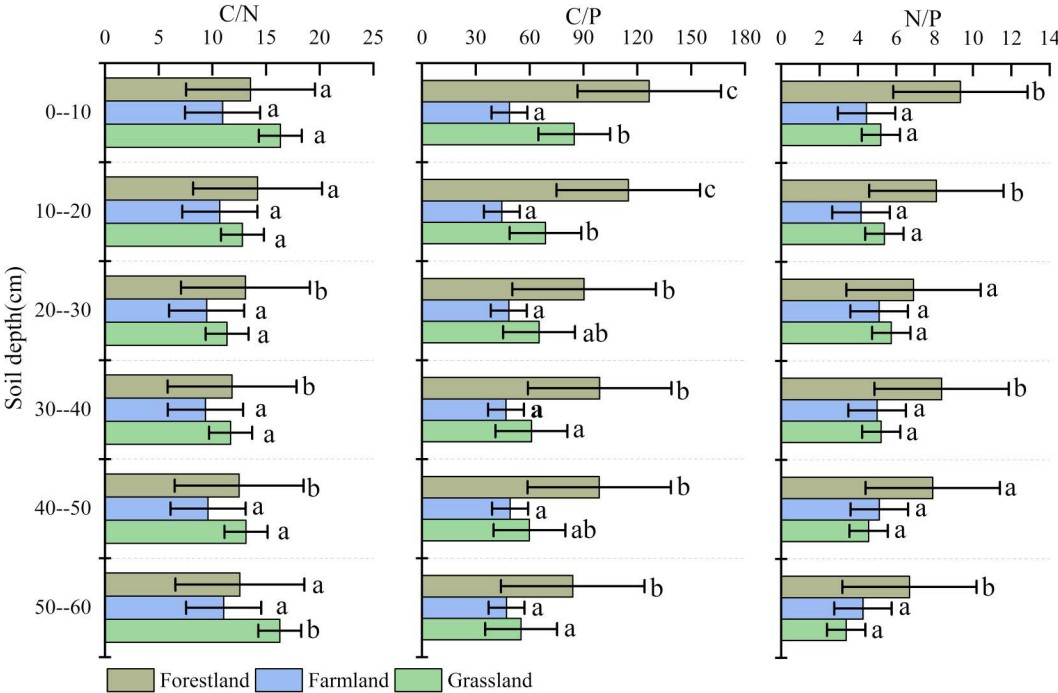

**Fig 6. The ecological stoichiometry of soil in vertical feature.** (Different small letters mean a significant difference at the 0.05 level).

and was higher than that in grassland and farmland overall. The N:P ratio of both farmland and grassland exhibited a "first increase and then decrease" pattern, with maximum values of 5.11 and 5.73 occurring in the 20–30 cm soil layer, respectively.

Plant and animal residues mainly accumulate on the soil surface, and soil microorganisms are more active in the shallow layer with strong decomposition and phosphorus mineralisation capacity [33,34], resulting in the SOC and TN contents being the largest in the surface layer and decreasing rapidly with the increase of the soil layer, whereas the soil TP showed relatively stable characteristics, with insignificant changes with the increase of soil layers. As a result, the C:N to C:P ratio in the top soil layer (0–30 cm) gradually decreased.

As previously described, with the increase in soil depth, the input of organic matter gradually decreases, microbial activity reduces, and decomposition weakens. The contents of SOC and TN gradually decrease with the increase in soil layers (with no obvious changes in cropland). However, the TP content in different soil layers of grassland and cropland is relatively stable, showing no obvious changes with the increase in soil layers. Although the TP content in forestland soil also shows a decreasing trend with the increase in soil layers, the rate of its decrease is lower than that of SOC and TN. This should be the main reason leading to the maximum C:P and N:P ratios in the surface soil of forestland, and the C:P and N:P ratios in the 0–30 cm soil layer of grassland soil being both greater than those in the 30–60 cm soil layer.

This indicates that the surface soil exhibits a strong nitrogen mineralization capacity. As decomposition progresses, easily degradable material is consumed, and nitrogen is immobilized in microbial biomass and decay products. This leaves behind more recalcitrant material, which decomposes at slower rates [35].

Land use changes had significant effects on the ecological stoichiometric characteristics of soil carbon, nitrogen and phosphorus (Fig 6). Specifically, there were significant differences *(P<0.05)* in C/N between different land use modes in all soil layers except 0–10 cm and 10–20 cm; C/P showed significant differences *(P<0.05)* between different land use

modes as a whole; and there were also significant differences *(P<0.05)* in N/P between different land use modes in all layers except 20–30 cm and 40–50 cm soil layers. significant differences *(P<0.05)*.

## 4. Discussion

### 4.1. Temperature and precipitation affect the annual variation of soil carbon, nitrogen and phosphorus

The decomposition and mineralization of plant litter are crucial processes in soil nutrient cycling and energy flow. Climate influences the decomposition and mineralization of plant litter by affecting microbial activity. Generally, the decomposition rate of soil microorganisms increases with rising air temperature (0–35°C), while precipitation influences the decomposition process by affecting leaching and the activity of decomposing agents [36]. From an annual variation perspective, the response of forestland and grassland to climate change is clear: In spring (May), lower temperature and reduced precipitation limit soil temperature and moisture, which in turn restricts microbial activity and weakens decomposition. In summer (June-August), temperatures rise, reaching optimal levels for soil microorganism growth, thereby enhancing microbial activity. However, the carbon input from above-ground biomass into the soil is insufficient to support microbial activity under underground warming. Microorganisms will utilize the stored SOC, accelerating its consumption and enhancing soil nitrogen mineralization [37–39]. At the same time, increased precipitation raises soil moisture, causing soil aggregates to disperse and break apart, thereby releasing SOC and TN [40]. In autumn (September-October), lower temperature and precipitation reduced microbial decomposition of SOC and TN. Meanwhile, vegetation fades and enters the soil as plant residues, converting into soil organic matter, which increases SOC and TN content. Therefore, temperature, precipitation, SOC, and TN contents in grassland and forest land generally fluctuate inversely and synchronously, resulting in stable fluctuations in soil C:N, indicating a strong coupling between soil C and N. These elements do not vary significantly with environmental changes and respond consistently to the same factors [41]. However, we observed that soil TN content in the study area followed the same trend as temperature and precipitation (except in October), causing a significant change in the C:N ratio in grassland from spring to summer. This is likely due to increased human grazing activities in summer, which convert animal manure into organic nitrogen stored in the soil as ammonium nitrogen through microbial processes. Soil TP content is relatively stable, except for autumn forest soil TP. Therefore, the trends in C:P and N:P ratios align with the trends in SOC and TN content, respectively, and show no significant relationship with temperature and precipitation, consistent with the findings of Post et al. (1985) [35]. From an inter-annual perspective, forestland across the three land types is more sensitive to precipitation variation, especially in May and June (Fig 7). A decrease in monthly total precipitation limited microbial activity and reduced biodecomposition rates. Consequently, SOC and TN contents in forest soil in May and June 2018 were higher than in 2019, indicating that spruce growth in Qinghai is highly sensitive to precipitation, consistent with findings by Guo et al., 2015 and Gao et al., 2018 [42–44].

In conclusion, fluctuations in temperature and precipitation during the growing season has the most significant impact on soil organic matter content in forestland, followed by grasslands and farmland. The carbon sequestration capacity of is relatively weak, mainly due to the lack of sufficient litter. Secondly, the destruction of soil aggregates and the enhancement of soil ventilation capacity also weaken the carbon sequestration capacity of soil [45,46]. This leads to a weakened carbon sequestration capacity and an imbalanced stoichiometric ratio.

### 4.2. Impacts of land use on soil carbon, nitrogen and phosphorus

Soil nutrient content was strongly influenced by land use. SOC content was primarily influenced by organic matter, including plant litter, microbial residues, and root exudates [26,47,48]. After the conversion of forest or grassland to agricultural land, litter quantity decreases, reducing carbon input from aboveground biomass to the soil. Additionally, reclamation improves soil aeration, exposes organic matter to the air, enhances soil temperature and moisture conditions, increases aerobic microbial activity, stimulates soil respiration, and accelerates the mineralization and decomposition of soil organic carbon [13,49,50], leading to reduced SOC content. Meanwhile, forestland soils promote the formation of large

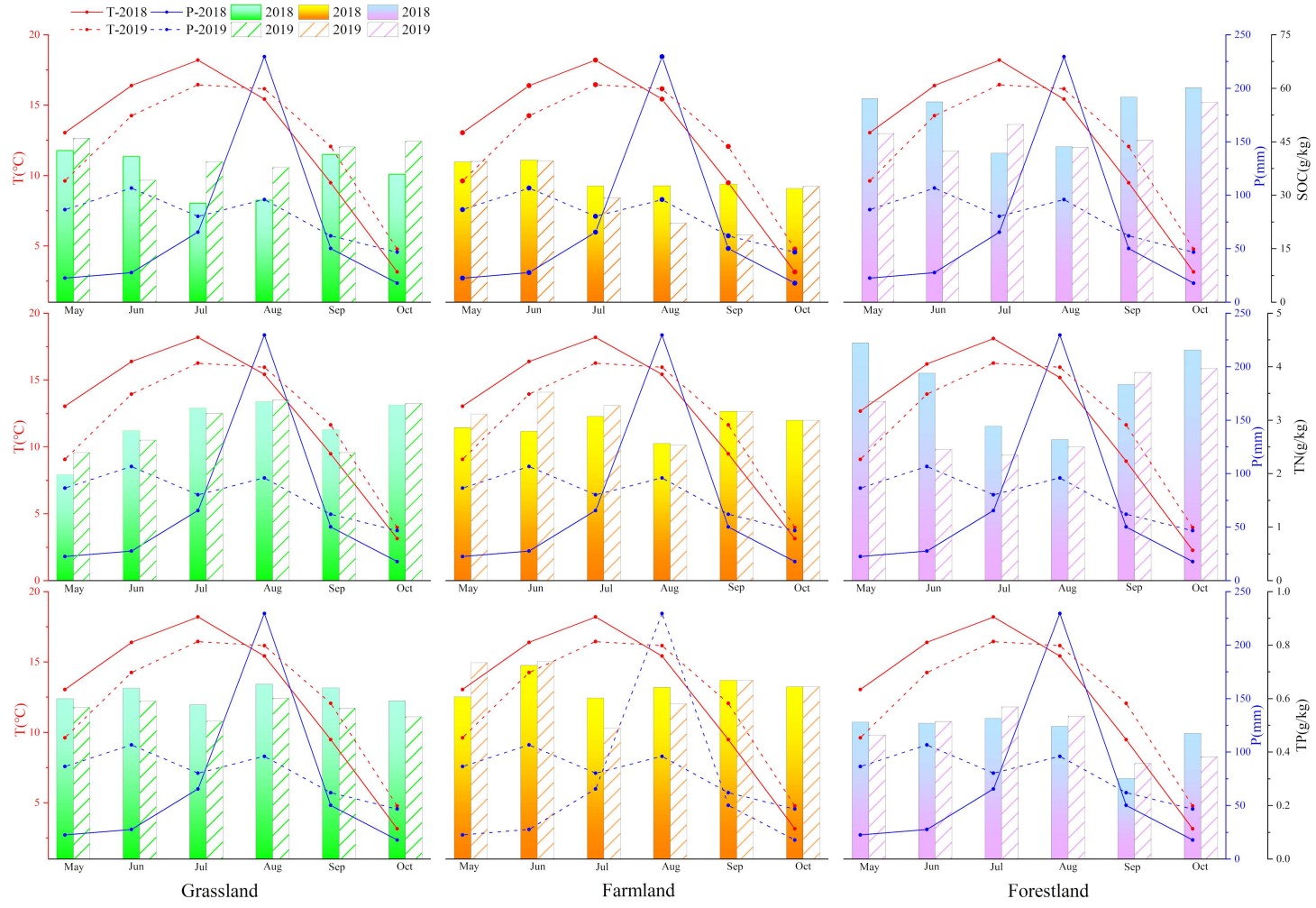

**Fig 7. The relationship between temperature, precipitation and SOC, TN and TP contents in different land use types.**

aggregates, which protect SOC from microbial decomposition [40,51]. Therefore, this study found significant differences in SOC content among forest land, grassland, and farmland, with forest and grassland soils having much higher SOC content than agricultural land (Table 1), consistent with previous studies [52].

N is one of the most important indicators of soil nutrients. Approximately 95% of soil nitrogen is derived from soil organic matter [53]. In natural forest and grassland ecosystems, TN is primarily derived from the decomposition of plant and animal residues. With increasing soil depth, reduced organic inputs result in stronger nitrogen mineralization in surface layers. As decomposition proceeds, labile organic matter is depleted, and nitrogen is sequestered in microbial biomass and decomposition byproducts. This leaves recalcitrant organic materials that decompose at slower rates [35], leading to a gradual decline in soil TN from the surface to deeper layers—ultimately stabilizing at 40–60 cm depth in natural vegetation soils. Nitrogen loss occurs through a process similar to carbon loss. As organic matter decreases, soil nitrogen also gradually declines. However, light grazing slightly increases root and soil nitrogen content [54] and soil C:N ratio [55] Although the soil nitrogen content in grasslands is slightly lower than that in forests, the difference is small (Table 1). Grassland soils, especially in summer (Fig 5), have relatively high nitrogen content. The slightly higher nitrogen content in cultivated soils compared to grasslands is primarily due to the application of urea and ammonia nitrogen fertilizers.

**Table 1. Soil carbon, nitrogen and phosphorus contents and ecological stoichiometric ratios of different land use types. Values are average±-standard error.**

| | SOC (g/kg) | TN (g/kg) | TP (g/kg) | C/N | C/P | N/P |
|---|---|---|---|---|---|---|
| Forestland | 48.82±7.59 | 3.52±0.77 | 0.46±0.09 | 13.87±6.11 | 106.13±48.82 | 7.65±3.22 |
| Grassland | 38.32±5.87 | 2.85±0.44 | 0.58±0.04 | 13.45±6.54 | 66.07±11.80 | 4.91±0.98 |
| Farmland | 30.8±5.94 | 3.04±0.25 | 0.65±0.07 | 10.13±1.98 | 47.38±8.32 | 4.67±0.79 |

The lower the C:N ratio, the faster nitrogen is released into the soil for crop use [29]. The C:N ratio of farmland fluctuated around 10.18, and the C:N ratio of soil surface organic matter under conventional tillage ranged from 8 to 15, with an average of 10–12 [32]. In this paper, the soil C:N ratio of arable land was 10.13±1.98, which was lower than that of grassland and forestland (Table 1).

P is an element with limited mobility [56]. A significant portion of soil phosphorus originates from the weathering of rocks and is primarily governed by geochemical processes [57]. Different land use patterns and cover types alter the phosphorus concentration in decomposed organic matter [58]. Grasses typically form extensive fine-root systems, with root biomass being a major contributor to nutrient input in grasslands. In contrast, root production and turnover rates in forestland are generally lower than in grasslands [59]. Agricultural practices can influence microbial community diversity and enhance phosphate mineralization capacity [60]. As a result, the total phosphorus (TP) content in cultivated soils significantly differs from that in forest and grassland soils. Plant and animal residues mainly accumulate on the soil surface, and soil microorganisms are more active in the shallow layer with strong decomposition and phosphorus mineralisation capacity [34,61]. This resulted in higher TP levels in the surface soils of forestland, grassland and arable land than in the middle and deeper layers. The soil C:P ratio is an important indicator of phosphorus mineralization capacity and a potential measure of soil organic matter's ability to release or absorb phosphorus from the environment [30]. Across the three land use types, forest soils exhibited higher C:P ratios than grasslands and croplands, peaking at 190.88 in autumn. Such elevated C:P ratios reduce soil phosphorus availability, thereby inhibiting phosphorus uptake by forest vegetation—a finding that demands careful consideration in future ecological management strategies.

Therefore, when land use changes, nutrients should be added to different soil types, taking into full account the limitations of C, N, and P elements in each land use/cover type.

### 4.3. Uncertainty and limitations

In this study, continuous sampling was conducted during the 2018–2019 growing seasons to compare the seasonal variation characteristics of soil carbon, nitrogen, and phosphorus in different land use types in the Beigouhe watershed on the northern slope of the eastern Qilian Mountains. This research provides an initial understanding of the impact of land use type changes and seasonal climate fluctuations on soil carbon, nitrogen, and phosphorus. However, the sampling was limited to the 2018–2019 growing seasons and a small part of the Beigouhe watershed, with a short time series and limited spatial coverage. Additionally, due to seasonal frozen soil, the absence of samples from non-growing seasons may have hindered capturing the full spatial diversity and seasonal changes of soil carbon, nitrogen, and phosphorus in the region. Meanwhile, this study attempts to analyze the impact of seasonal climate fluctuations (temperature and precipitation changes) on the stability of soil carbon, nitrogen, and phosphorus during the growing season (May-October), but due to the lack of non-growing season data and soil samples from larger spatial scales, the effects of asymmetric seasonal or spatial warming on land use types and soil nutrient stability could not be fully discussed. Therefore, future work will focus on increasing soil sampling in non-growing seasons and expanding the spatial coverage of soil samples to further strengthen the research on the spatial diversity and seasonal variations of soil carbon, nitrogen, and phosphorus in the Beigouhe watershed.

## 5. Conclusions

This paper analysed the spatial and temporal changes of soil carbon, nitrogen and phosphorus contents and their ecological stoichiometric ratios in, grassland and cropland soils in the Binggou River Basin in the eastern section of the Qilian Mountains based on continuous field sampling data. The effects of land use changes and seasonal climate fluctuations on soil carbon, nitrogen and phosphorus were discussed. The following main conclusions were obtained:

The SOC content of forestland soil was significantly higher than that of grassland and cropland, and the conversion of forestland and grassland into cropland decreased the SOC content by 16.04 g/kg and 4.8 g/kg, respectively. This decreased the soil carbon sequestration capacity significantly and increased the risk of soil carbon emission. Thus, in the future ecological restoration process, efforts to return ploughland to forest and grassland should be further increased to enhance the ecosystem service function of the Qilian Mountain area. The TP content of forest soil was significantly lower than that of grassland and cropland, and the difference in TN content was very small, resulting in significantly higher C:P and N:P ratios in forest soil than in grassland and cropland. Especially in autumn, the C:P and N:P ratios in forest soil were much higher than those in other seasons. Excessive C:P reduced the effectiveness of soil phosphorus. The weaker carbon sequestration capacity of arable land resulted in significantly lower C:N (10.13) and C:P (47.38) ratios than those of forestland and grassland. Lower C:N and C:P ratios weakened soil microbial activity, hindered nutrient uptake by plants, and inhibited root growth. In terms of seasonal changes, the SOC content of the three land use types showed the characteristic of "low in summer and high in spring and autumn", while the TN content and C:N ratio of grassland soil were higher in summer and lower in spring and autumn, and the autumn TN content showed seasonal fluctuations contrary to those of. In the soil vertical profile, the contents of SOC, TN, and TP in forest land showed a significant decreasing trend with the increase of the soil layer, especially in the 0–30 cm soil layer. The characteristics of changes in SOC and TN in the vertical profile of grassland soil were similar to those of, but the maximum value of soil TP content appeared in the 50–60 soil layer. The three kinds of nutrients in arable soils showed an "S" type of change in the vertical direction, and the reduction of tillage, ploughing, and the use of chemical fertilisers changed the distribution of soil nutrients in the vertical profile of the soils. In the future ecological restoration process, attention should be paid to the problem of low TP content and high C:P in autumn. For arable land that still existed, measures such as returning straw to the field and applying organic fertilisers were taken to increase the input of organic matter, enhance the carbon sequestration capacity of arable land soils, improve soil fertility, prevent soil degradation, and promote plant growth.

Based on the existence of seasonal permafrost in the study area, this paper conducted continuous sampling from May to October in the plant growing season of 2018–2019, carried out a comparative study of soil carbon, nitrogen, and phosphorus characteristics of different land-use types in the Binggou River Basin on the northern slope of the eastern section of the Qilian Mountains, and preliminarily grasped the laws of land-use type changes and seasonal climatic fluctuations on soil carbon, nitrogen, and phosphorus. However, since the sample collection was limited to the plant growing season and localised areas in the Binggou River Basin, with a short time series, small sampling area, and lack of sampling data from non-growing seasons, it might not fully capture the spatial diversity and seasonal variation characteristics of regional soil carbon, nitrogen, and phosphorus. It was not possible to discuss the effects of asymmetric seasonal or spatial warming on land use types and soil nutrient stability more fully. Therefore, in future work, the collection of soil samples in the non-growing season should be increased, the spatial range of soil sample collection should be expanded, and the time series of sample collection should be lengthened to further study the spatial and temporal variation of soil carbon, nitrogen, and phosphorus in the Binggou River Basin in depth, as well as its driving mechanism.

## Supporting information

**S1 File. Minimum data set used in the article.**
(XLS)

**S2 File. Figs 1–7 is an illustration from the manuscript; Table 1 is Soil carbon, nitrogen and phosphorus contents and ecological stoichiometric ratios of different land use types.**
(DOCX)

## Author contributions

**Conceptualization:** Junju Zhou, Shizhen Xu, Wei Shi, Dongxia Zhang, Guofeng Zhu, Xuemei Yang, Chunli Wang, Wei Wei, Haiyan Ma.

**Data curation:** Haiyan Ma.

**Funding acquisition:** Junju Zhou, Guofeng Zhu, Xuemei Yang.

**Methodology:** Junju Zhou, Shizhen Xu, Haihua Shan, Bingxing Li, Wei Wei.

**Software:** Junju Zhou, Shizhen Xu, Haihua Shan, Bingxing Li, Wei Shi, Dongxia Zhang, Chunli Wang, Wei Wei.

**Writing – original draft:** Junju Zhou, Shizhen Xu.

**Writing – review & editing:** Junju Zhou, Shizhen Xu.

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
