## [Decision Letter · Decision Letter 0]

PONE-D-24-59609Impacts of Land Use on Soil Carbon, Nitrogen, and Phosphorus Cycling in the Eastern Qilian MountainsPLOS ONE

Dear Dr. Zhou,

Thank you for submitting your manuscript to PLOS ONE. After careful consideration, we feel that it has merit but does not fully meet PLOS ONE’s publication criteria as it currently stands. Therefore, we invite you to submit a revised version of the manuscript that addresses the points raised during the review process.

**ACADEMIC EDITOR:** The study has value but the manuscript has some problems as suggested by the reviewers. I read the manuscript and the comments of the three reviewers. In my opinion, the manuscript might be improved by a major revision. The authors should respond to the comments of the reviewers one by one and revise the manuscript accordingly. The revised manuscript would be sent to the reviewers for further reviewing.

We look forward to receiving your revised manuscript.

Kind regards,

Jian Liu

Academic Editor

PLOS ONE

Journal Requirements:

This work was supported by National Natural Science Foundation of China (Grant No. 42361005, No. 41867030, No.32060373) for their financial support. 

5. Please amend your list of authors on the manuscript to ensure that each author is linked to an affiliation. Authors’ affiliations should reflect the institution where the work was done (if authors moved subsequently, you can also list the new affiliation stating “current affiliation:….” as necessary).

6. We note that Figure 1 in your submission contain [map/satellite] images which may be copyrighted. All PLOS content is published under the Creative Commons Attribution License (CC BY 4.0), which means that the manuscript, images, and Supporting Information files will be freely available online, and any third party is permitted to access, download, copy, distribute, and use these materials in any way, even commercially, with proper attribution. For these reasons, we cannot publish previously copyrighted maps or satellite images created using proprietary data, such as Google software (Google Maps, Street View, and Earth). For more information, see our copyright guidelines: http://journals.plos.org/plosone/s/licenses-and-copyright.

7. Please ensure that you refer to Figure 7 in your text as, if accepted, production will need this reference to link the reader to the figure.

8. We note you have included a table to which you do not refer in the text of your manuscript. Please ensure that you refer to Table 1 in your text; if accepted, production will need this reference to link the reader to the Table.

Reviewers' comments:

Reviewer's Responses to Questions

**Comments to the Author**

1. Is the manuscript technically sound, and do the data support the conclusions?

Reviewer #1: Yes

Reviewer #2: Yes

Reviewer #3: Partly

2. Has the statistical analysis been performed appropriately and rigorously? 

Reviewer #1: Yes

Reviewer #2: No

Reviewer #3: No

3. Have the authors made all data underlying the findings in their manuscript fully available?

Reviewer #1: Yes

Reviewer #2: No

Reviewer #3: No

4. Is the manuscript presented in an intelligible fashion and written in standard English?

Reviewer #1: Yes

Reviewer #2: No

Reviewer #3: Yes

5. Review Comments to the Author

Reviewer #1: General Comments

This study investigates the impact of land use on the cycling of soil carbon (C), nitrogen (N), and phosphorus (P) in the Binggou River Basin of the Eastern Qilian Mountains, focusing on three land types: forest, grassland, and cultivated land. The study demonstrates that land use changes influence the ecological stoichiometry of these elements, showing that conversion from forest or grassland to cultivated land reduces soil organic carbon (SOC) and nitrogen content, causing the land to shift from a "carbon sink" to a "carbon source." The study further highlights that soil nutrient availability varies across different land types, with phosphorus becoming a limiting nutrient in forest soils, particularly in autumn. The study suggests strategies such as forest and grassland restoration, enhanced carbon sequestration, and reduced use of agricultural fertilizers to promote soil health and nutrient balance, providing theoretical support for ecological restoration. This paper is highly worthy of publication. However, before formal acceptance, it is recommended that some revisions be made.

Main Comments

Although the study includes a large sample size, the sampling period is limited to 2018-2019, and the data are primarily concentrated in a small geographical area. Therefore, the spatial diversity and seasonal variation of the region may not be fully captured. It is recommended to add a discussion on uncertainty.

While the discussion section addresses the impact of climate factors on soil nutrients, it does not sufficiently explore the long-term effects of different climate change scenarios on soil nutrient dynamics, such as asymmetric seasonal or spatial warming, etc (doi: 10.3389/fpls.2023.1090204; doi: 10.3389/fevo.2021.757943). It is recommended to add relevant discussions.

Minor Comments

1. Line 456: "conficts of interest" should be corrected to "conflicts of interest."

2. Line 456: "exits" should be corrected to "exists."

3. "microbial biomass phosphorus" should be revised to "microbial biomasses phosphorus."

4. Figure 2 could benefit from using monthly data to match the other figures.

5. Figure 4 lacks an x-axis label and should be corrected.

6. "This study analyzes" should be changed to "This study analyzed."

7. Line 287: "The results show" should be changed to "The results showed."

8. The sentence "From 2018 to 2019, the average soil organic carbon (SOC) content in forest land (48.82 g/kg) was higher than in grassland and cultivated land" should be revised to: "From 2018 to 2019, the average soil organic carbon (SOC) content in forest land (48.82 g/kg) was found to be higher than that in grassland and cultivated land" to improve clarity.

9. The color scheme in Figure 6 appears somewhat jarring and may need adjustment.

10. The references section has some formatting issues, such as incorrect indentation (e.g., line 474).

Reviewer #2: Impacts of land use on soil carbon, nitrogen, and phosphorus cycling in the eastern Qilian mountains

This manuscript explores the spatial and temporal patterns in forest, grassland, and cultivated land and presents some interesting results. However, further modifications and improvements are necessary to enhance its clarity and rigor.

Regarding theme of the manuscript, which focuses on different land use, the relationship between forest, grassland and cultivated land remains unclear. Line 95-96 mentions a series of ecological restoration projects have been implemented in Qilian Mountains since 2000. What about the three samples? According to Line 123-124 and Line 215-216, the grassland and woodland have been converted into cultivated land. There are some contradiction and confusion for me. Therefore, the authors should provide to clarify the relationship between these samples.

Furthermore, soil carbon, nitrogen and phosphorus cycling typically involve various processes, but this manuscript only considers soil organic carbon, nitrogen and phosphorus. Therefore, the title of the manuscript may need further consideration.

L14. The terms “forest” and “woodland” are used interchangeably in the manuscript. For consistency, the authors should choose one term and use it throughout the document.

L27. The statement that lower C:N and C:P ratios weaken soil microorganisms, hindering plant nutrient absorption and inhibiting root growth, is not supported by data. Therefore, the authors should revise the statement or cite references to be more accurate.

L31. The conclusion that phosphorus becomes the limiting nutrient for forest growth in autumn cannot be directly inferred from the data presented. The authors should provide additional evidence or revise this statement to be more accurate.

L152, L179, L194. These statements mention the soil particle size of soil for the measurement of TN and TP, but there is a contradiction between 0.25 mm and 100-mesh sieve.

L192, L207. Equipment used in the manuscript should be marked with the model, manufacturer and country.

L209-L210. Significance test for different land use, seasonal and so on are missing.

L274-275, L284-286, L339-342. These sentences, which serve as argument, should be included in the discussion section of the manuscript for better organization and clarity.

L308-309, L330-331, L336-338. The sentence should be convincing evidence, based on the references or the data, to support these statements.

L380. Climate change refers to changes in the climate over a long period of time. The data of manuscript just only involves from 2018 to 2019, then how to get the conclusion “climate change has the most significant impact on soil organic matter content in woodlands, followed by grasslands and cultivated lands”?

L382-383. In the manuscript, cultivated land through tillage, disrupts soil aggregates and increases ventilation lead to a weakened carbon sequestration capacity and an imbalanced stoichiometric ratio. How to get the result? Isn’t the lack of litter in cultivated land as a more important reason for the pattern of soil organic carbon?

L419. Please provide the description of table1. How to get the parameter of ecostoichiometric ratio?

The conclusion section is a summary of the research results, as well as an in-depth analysis of the research achievements and a prospect for future studies. It consolidates and refines these facts based on the research outcomes, while maintaining rigor by avoiding over-extrapolation (L434, L443, L446). The conclusion requires revisiting and reorganization.

Reviewer #3: Taking the Binggou River Basin (north slope of the eastern Qilian Mountains), as study area, a sensitive area to global change and play a crucial role in maintaining regional ecological security as a national nature reserve. I had comprehensively and deeply reviewed the manuscript, which was entitled: The Impacts of Land Use on Soil Carbon, Nitrogen, and Phosphorus Cycling in the Eastern Qilian Mountains. This is a topic of some interest to researchers in related fields, but the paper needs a lot of improvement before it can be accepted. My detailed comments are as follows:

1. “Soil Carbon, Nitrogen, and Phosphorus Cycling”, How is it defined? How is this measured in your experimental design? The title of the manuscript does not correspond well with the research content.

2. In terms of experimental design, only one plot is set for each land use type, and there is no duplication in land use. Can it support your research conclusions.

3. Do the three sampling plots of different types of land use have the same/similar conditions in terms of terrain, light, soil, etc.?

4. In terms of data analysis, ANOVA and significance test are lacking.

5. The innovation of the research is not clearly expressed.

6. PLOS authors have the option to publish the peer review history of their article (what does this mean? ). If published, this will include your full peer review and any attached files.

**Do you want your identity to be public for this peer review?** For information about this choice, including consent withdrawal, please see our Privacy Policy .

Reviewer #1: No

Reviewer #2: No

Reviewer #3: No

---

## [Author Response · Author response to Decision Letter 1]

8 Apr 2025

Dear Editor and Reviewers:

We are grateful for your consideration of our manuscript entitled "Impacts of Land Use on Soil Carbon, Nitrogen, and Phosphorus Cycling in the Eastern Qilian Mountains" [PONE-D-24-59609], and we also very much appreciate your constructive comments and useful suggestions, which have enabled us to improve the manuscript. All the comments we received on this study have been considered, and we present our reply to each of them separately. We hope the revised manuscript will satisfy you.The following is an illustration of the different coloured revisions in the manuscript with markings: blue with strikethrough are markings removed from the original text, yellow highlights are reworked additions, and red headings are reworked sections.

Reviewer #1: General Comments

This study investigates the impact of land use on the cycling of soil carbon (C), nitrogen (N), and phosphorus (P) in the Binggou River Basin of the Eastern Qilian Mountains, focusing on three land types: forest, grassland, and cultivated land. The study demonstrates that land use changes influence the ecological stoichiometry of these elements, showing that conversion from forest or grassland to cultivated land reduces soil organic carbon (SOC) and nitrogen content, causing the land to shift from a "carbon sink" to a "carbon source." The study further highlights that soil nutrient availability varies across different land types, with phosphorus becoming a limiting nutrient in forest soils, particularly in autumn. The study suggests strategies such as forest and grassland restoration, enhanced carbon sequestration, and reduced use of agricultural fertilizers to promote soil health and nutrient balance, providing theoretical support for ecological restoration. This paper is highly worthy of publication. However, before formal acceptance, it is recommended that some revisions be made.

Main Comments

Although the study includes a large sample size, the sampling period is limited to 2018-2019, and the data are primarily concentrated in a small geographical area. Therefore, the spatial diversity and seasonal variation of the region may not be fully captured. It is recommended to add a discussion on uncertainty.

While the discussion section addresses the impact of climate factors on soil nutrients, it does not sufficiently explore the long-term effects of different climate change scenarios on soil nutrient dynamics, such as asymmetric seasonal or spatial warming, etc (doi: 10.3389/fpls.2023.1090204; doi: 10.3389/fevo.2021.757943). It is recommended to add relevant discussions.

Response: Thank you for your suggestion. We have added a discussion on uncertainties in the discussion section (4.3) (Line 534-551), and the references have been cited as follows (doi:10.3389/fpls.2023.1090204; doi:10.3389/fevo.2021.757943). The specific content is as follows:

In this study, continuous sampling was conducted during the 2018-2019 growing seasons to compare the seasonal variation characteristics of soil carbon, nitrogen, and phosphorus in different land use types in the Beigouhe watershed on the northern slope of the eastern Qilian Mountains. This research provides an initial understanding of the impact of land use type changes and seasonal climate fluctuations on soil carbon, nitrogen, and phosphorus. However, the sampling was limited to the 2018-2019 growing seasons and a small part of the Beigouhe watershed, with a short time series and limited spatial coverage. Additionally, due to seasonal frozen soil, the absence of samples from non-growing seasons may have hindered capturing the full spatial diversity and seasonal changes of soil carbon, nitrogen, and phosphorus in the region. Meanwhile, this study attempts to analyze the impact of seasonal climate fluctuations (temperature and precipitation changes) on the stability of soil carbon, nitrogen, and phosphorus during the growing season (May-October), but due to the lack of non-growing season data and soil samples from larger spatial scales, the effects of asymmetric seasonal or spatial warming on land use types and soil nutrient stability could not be fully discussed (Jiangwei Wang, et al., 2021; Fusong Han, et al., 2023). Therefore, future work will focus on increasing soil sampling in non-growing seasons and expanding the spatial coverage of soil samples to further strengthen the research on the spatial diversity and seasonal variations of soil carbon, nitrogen, and phosphorus in the Beigouhe watershed.

Minor Comments

Response: Thank you for your kind comments and suggestions.We have carefully revised the manuscript based on your feedback and sincerely hope that the revised version meets your expectations.

1. Line 456: "conficts of interest" should be corrected to "conflicts of interest."

Response 1: Thank you very much for your valuable suggestions. We have corrected "conficts of interest" to "conflicts of interest."(Line 610)

2.Line 456: "exits" should be corrected to "exists."

Response 2: Thank you very much for your valuable suggestions. We have corrected "exits" to "exists."

3. "microbial biomass phosphorus" should be revised to "microbial biomasses phosphorus."

Response 3: Thank you very much for your valuable suggestions. We have corrected "microbial biomass phosphorus" to "microbial biomasses phosphorus."

4. Figure 2 could benefit from using monthly data to match the other figures.

Response 4: Thank you very much for your valuable suggestions. We have modified the data in the figure to monthly data.

5. Figure 4 lacks an x-axis label and should be corrected.

Response 5: Thank you very much for your valuable suggestions. We have added the missing x-axis label to Figure 4 and have redrawn the figure accordingly.

6. "This study analyzes" should be changed to "This study analyzed."

Response 6: Thank you very much for your valuable suggestions. We have corrected "This study analyzes" to "This study analyzed."

7. Line 287: "The results show" should be changed to "The results showed."

Response 7: Thank you very much for your valuable suggestions. We have corrected "The results show" to "The results showed."

8. The sentence "From 2018 to 2019, the average soil organic carbon (SOC) content in forest land (48.82 g/kg) was higher than in grassland and cultivated land" should be revised to: "From 2018 to 2019, the average soil organic carbon (SOC) content in forest land (48.82 g/kg) was found to be higher than that in grassland and cultivated land" to improve clarity.

Response 8: Thank you very much for your valuable suggestions. We have corrected "From 2018 to 2019, the average soil organic carbon (SOC) content in forest land (48.82 g/kg) was higher than in grassland and cultivated land" to "From 2018 to 2019, the average soil organic carbon (SOC) content in forest land (48.82 g/kg) was found to be higher than that in grassland and cultivated land."

9. The color scheme in Figure 6 appears somewhat jarring and may need adjustment.

Response 9: Thank you very much for your valuable suggestions. We have adjusted the color scheme in Figure 6 and redrawn the figure accordingly.

10. The references section has some formatting issues, such as incorrect indentation (e.g., line 474).

Response 10: Thank you very much for your valuable suggestions. We have revised the format of the references as required.

Reviewer #2: Impacts of land use on soil carbon, nitrogen, and phosphorus cycling in the eastern Qilian mountains. This manuscript explores the spatial and temporal patterns in forest, grassland, and cultivated land and presents some interesting results. However, further modifications and improvements are necessary to enhance its clarity and rigor.

Regarding theme of the manuscript, which focuses on different land use, the relationship between forest, grassland and cultivated land remains unclear. Line 95-96 mentions a series of ecological restoration projects have been implemented in Qilian Mountains since 2000. What about the three samples? According to Line 123-124 and Line 215-216, the grassland and woodland have been converted into cultivated land. There are some contradiction and confusion for me. Therefore, the authors should provide to clarify the relationship between these samples.

Response: Thank you for your questions and suggestions. We acknowledge that the expression of this issue was indeed unclear. Based on your suggestion, we have made revisions (Line 105-113).

In recent decades, global climate change has had a significant impact on the Qilian Mountain ecosystem. Additionally, the expansion of cultivated land and the reclamation of forests and grasslands by local residents have further degraded the ecosystem. Although China has implemented a series of ecological restoration projects since 2000, including natural forest protection, the construction of ecological public welfare forests, and the return of farmland to forest (or grassland) to protect and restore the degraded ecosystem (Wang et al., 2019), some of the reclaimed farmland has yet to be converted back to grassland or forest. Changes in land use types have altered ecosystem services and impacted the processes of soil nutrient retention and cycling.

Furthermore, soil carbon, nitrogen and phosphorus cycling typically involve various processes, but this manuscript only considers soil organic carbon, nitrogen and phosphorus. Therefore, the title of the manuscript may need further consideration.

Response: Thank you for your suggestions. After careful consideration and in response to your feedback, we have revised the title of the paper.

The title "Impacts of Land Use on Soil Carbon, Nitrogen, and Phosphorus Cycling in the Eastern Qilian Mountains" has been changed to "Impacts of Land Use on Soil Carbon, Nitrogen, and Phosphorus in the Eastern Qilian Mountains."

L14. The terms “forest” and “woodland” are used interchangeably in the manuscript. For consistency, the authors should choose one term and use it throughout the document.

Response: Thank you for your suggestion. We have made the revision and have standardized the use of "forest land" throughout the manuscript.

L27. The statement that lower C:N and C:P ratios weaken soil microorganisms, hindering plant nutrient absorption and inhibiting root growth, is not supported by data. Therefore, the authors should revise the statement or cite references to be more accurate.

Response: Thank you for your suggestion. It is true that we did not test soil microorganisms, and that conclusion is a combination of existing research results (Du et al., 2020). , which was inferred from the C:N and C:P characteristics of the arable soils in this paper. We removed this sentence based on insufficient data to support this conclusion.

Du, E., Terrer, C., Pellegrini, A.F.A. et al. Global patterns of terrestrial nitrogen and phosphorus limitation. Nat. Geosci. 13, 221–226 (2020). https://doi.org/10.1038/s41561-019-0530-4

L31. The conclusion that phosphorus becomes the limiting nutrient for forest growth in autumn cannot be directly inferred from the data presented. The authors should provide additional evidence or revise this statement to be more accurate.

Response: Thank you for your questions and suggestions. We have revised this paragraph of the summary as a whole

L152, L179, L194. These statements mention the soil particle size of soil for the measurement of TN and TP, but there is a contradiction between 0.25 mm and 100-mesh sieve.

Response: Thank you for your questions and suggestions. Soil samples were air-dried for a fortnight in a cool, ventilated environment. Plant debris and gravel were removed from the soil samples prior to soil splitting. The air-dried soil samples were ground and passed through a 100 mesh sieve to remove coarse fragments and debris for the determination of soil organic carbon (SOC), soil total nitrogen (TN), and soil total phosphorus (TP).

L192, L207. Equipment used in the manuscript should be marked with the model, manufacturer and country.

Response: Thank you for your questions and suggestions. We have revised the original text as follows:Soil sample measurements were carried out using the Smartchem 200 Discrete Auto Analyzer, branded AMS Alliance, Italy.

L209-L210. Significance test for different land use, seasonal and so on are missing.

Response:Thank you for your suggestions. We have added the method for testing the significance level and conducted tests on the differences in carbon, nitrogen, and phosphorus content across different land use types, as well as the significance of seasonal variations.

L274-275, L284-286, L339-342. These sentences, which serve as argument, should be included in the discussion section of the manuscript for better organization and clarity.

Response: Thank you for your questions and suggestions. We have incorporated these three sentences into the discussion section to better support our research, respectively in the subsections.

L308-309, L330-331, L336-338. The sentence should be convincing evidence, based on the references or the data, to support these statements.

Response: Thank you for your suggestion, we have carefully reviewed these sentences and have deleted and modified them as follows, taking into account that they do not explain the data in the text very well:

L308-309

Response: Thanks to your suggestion. Based on the fact that during our survey of the study area we found that the grassland was grazed and that nitrogen fertiliser was used on the arable land, the following adjustments were made to the paragraph where L308-309 is located:

During the investigation of the study area, it was found that grazing existed in the grassland, and nitrogen fertiliser would be used in the arable farming process, which increased the input of nitrogen, but the carbon sequestration capacity of the grassland and the arable land was weaker (Fig. 3), and the consumption of nitrogen was limited by carbon, which couldn't be absorbed by the plants sufficiently, and it was accumulated in the soil, which resulted in the TN content of the soil of the grassland was higher than that of its month of July in summer, and the TN content of the soil of the arable land decreased, but to a lesser extent, in July (Fig. 3). month was reduced, but the reduction was small (Figure 3). On the other hand, the phosphorus content of grassland and arable soils decreased significantly in summer, especially in July, resulting in higher soil N:P ratios in summer.

L330-331:

Response: Thanks to your suggestion, we have amended the above sentence as follows

Plant and animal residues mainly accumulate on the soil surface, and soil microorganisms are more active in the shallow layer with strong decomposition and phosphorus mineralisation capacity (Wang et al., 2021; Carrasco-Espinosa et al., 2024), resulting in the soil SOC and TN contents being the largest in the surface layer and decreasing rapidly with the increase of the soil layer, whereas the soil TP showed relatively stable characteristics, with insignificant changes with the increase of soil layers. As a result, the C:N to C:P ratio in the top soil layer (0-30 cm) gradually decreased.

L336-338

Response: Thanks to your suggestion, we have amended the above sentence as follows

As previously described, with the increase in soil depth, the input of organic matter gradually decreases, microbial activity reduces, and decomposition weakens. The contents of soil SOC and TN gradually decrease with the increase in soil layers (with no obvious changes in cropland). However, the TP content in different soil layers of grassland and cropland is relatively stable, showing no obvious changes with the increase in soil layers. Although the TP content in forestland soil also shows a decreasing trend with the increase in soil layers, the rate of its decrease is lower than that of SOC and TN. This should be the main reason leading to the maximum C:P and N:P ratios in the surface soil of forestland, and the C:P and N:P ratios in the 0–30 cm soil layer of grassland soil being both greater than those in the 30–60 cm soil layer.

L380. Climate change refers to changes in the climate over a long period of time. The data of manuscript just only involves from 2018 to 2019, then how to get the conclusion “climate change has the most significant

---

## [Decision Letter · Decision Letter 1]

PONE-D-24-59609R1Impacts of Land Use on Soil Carbon, Nitrogen, and Phosphorus in the Eastern Qilian MountainsPLOS ONE

Dear Dr. Zhou,

Thank you for submitting your manuscript to PLOS ONE. After careful consideration, we feel that it has merit but does not fully meet PLOS ONE’s publication criteria as it currently stands. Therefore, we invite you to submit a revised version of the manuscript that addresses the points raised during the review process.

We look forward to receiving your revised manuscript.

Kind regards,

Jian Liu

Academic Editor

PLOS ONE

Journal Requirements:

Additional Editor Comments:

The revised version has been improved a lot.  But the manuscript still has some problems as suggested by the two reviewers. The authors should respond to the comments of the reviewers one by one and revise the manuscript accordingly.  

Reviewers' comments:

Reviewer's Responses to Questions

**Comments to the Author**

1. If the authors have adequately addressed your comments raised in a previous round of review and you feel that this manuscript is now acceptable for publication, you may indicate that here to bypass the “Comments to the Author” section, enter your conflict of interest statement in the “Confidential to Editor” section, and submit your "Accept" recommendation.

Reviewer #2: All comments have been addressed

Reviewer #3: All comments have been addressed

2. Is the manuscript technically sound, and do the data support the conclusions?

Reviewer #2: Yes

Reviewer #3: Yes

3. Has the statistical analysis been performed appropriately and rigorously? 

Reviewer #2: Yes

Reviewer #3: Yes

4. Have the authors made all data underlying the findings in their manuscript fully available?

Reviewer #2: Yes

Reviewer #3: Yes

5. Is the manuscript presented in an intelligible fashion and written in standard English?

Reviewer #2: Yes

Reviewer #3: Yes

6. Review Comments to the Author

Reviewer #2: The revised manuscript shows substantial improvement and now ready for minor revisions before formal acceptance. Please address the following points:

1. L182, L196: The capitalization of the initial letters in the title should be consistent throughout. Please standardize this.

2. L 250, L251, L266, L285, Fig.4 and Table 1: It is recommended to include variability indicators (±SD or ±SE) for the key indicators in the paper.

3. “p” in the manuscript: The letter “p” should be represented in lowercase and italicized.

4. L447. “soil SOC” is the abbreviation of “soil soil organic carbon”, please revise it.

5. L494. TN has already been defined in the previous text, so there is no need to define it again.

6. Please modify the references according to the format requirements.

Reviewer #3: This version of the manuscript responds to the comments of the three reviewers one by one, and the quality of the manuscript has been greatly improved, but more specific and comprehensive information needs to be provided in some details through minor revision. My main suggestions are as follows:

Firstly, in the section “2.1 Description of the Study Area”, It is suggested to supplement the land use and cover changes of the whole study area over the past 10 years, especially the plant composition and community structure characteristics of each sampling site and the tillage of agricultural land.

Secondly, It is proposed to supplement the hydrological connectivity status of the Binggou River Basin, including the interaction of surface runoff in its adjacent catchments.

Thirdly, Please state what damage has been done to the ecological environment in the study area and to what extent.

Others, (1) There is room for improvement in some series of results analysis, for example, in Fig 7, Columns filled with patterns and colors only indicate the meaning of grassland, farmland and forestland are not marked. (2) In terms of the content arrangement of this manuscript, the results and the discussion are separate. It is suggested that the discussion expressions quoting references in the results should be included in the discussion chapter. (3) The evidence (references) cited in the discussion section should have similar research premises to this study, please check. (4) When it comes to the significance level of the impact/difference, it is recommended to indicate a probability value, such as p < xxx.

7. PLOS authors have the option to publish the peer review history of their article (what does this mean? ). If published, this will include your full peer review and any attached files.

**Do you want your identity to be public for this peer review?** For information about this choice, including consent withdrawal, please see our Privacy Policy .

Reviewer #2: No

Reviewer #3: No

---

## [Author Response · Author response to Decision Letter 2]

18 May 2025

Dear Editor and Reviewers:

We are grateful for your consideration of our manuscript entitled "Impacts of Land Use on Soil Carbon, Nitrogen, and Phosphorus Cycling in the Eastern Qilian Mountains" [PONE-D-24-59609R1], and we also very much appreciate your constructive comments and useful suggestions, which have enabled us to improve the manuscript. All the comments we received on this study have been considered, and we present our reply to each of them separately. We hope the revised manuscript will satisfy you.The following is an illustration of the different coloured revisions in the manuscript with markings: blue with strikethrough are markings removed from the original text, yellow highlights are reworked additions.

---

## [Editor Report · Decision Letter 2]

Impacts of land use on soil carbon, nitrogen, and phosphorus in the Eastern Qilian Mountains

PONE-D-24-59609R2

Dear Dr. Zhou,

We’re pleased to inform you that your manuscript has been judged scientifically suitable for publication and will be formally accepted for publication once it meets all outstanding technical requirements.

Kind regards,

Jian Liu

Academic Editor

PLOS ONE
---

## [Editor Report · Acceptance letter]

PONE-D-24-59609R2

PLOS ONE

Dear Dr. Zhou,

I'm pleased to inform you that your manuscript has been deemed suitable for publication in PLOS ONE. Congratulations! Your manuscript is now being handed over to our production team.

Kind regards,

on behalf of

Dr. Jian Liu

Academic Editor

PLOS ONE